# GTR: Improving Large 3D Reconstruction Models through Geometry and Texture Refinement

**Peiye Zhuang**        **Songfang Han**        **Chaoyang Wang**        **Aliaksandr Siarohin**

**Jiaxu Zou**        **Michael Vasilkovsky**        **Vladislav Shakhrai**        **Sergei Korolev**

**Sergey Tulyakov**                    **Hsin-Ying Lee**

https://snap-research.github.io/GTR/

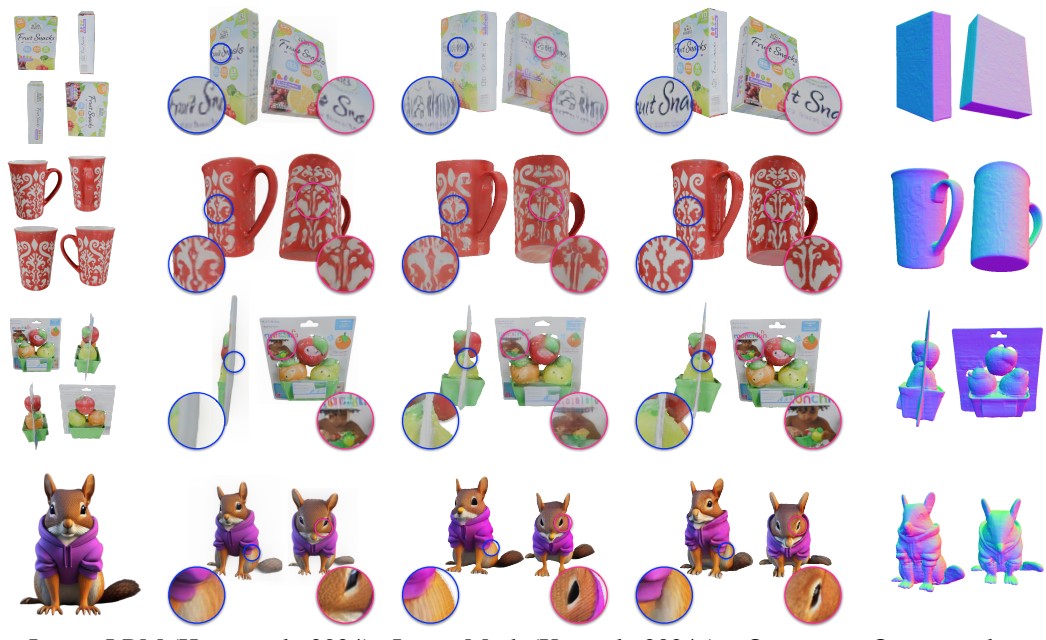

Figure 1: **Comparison with the baseline methods on the sparse-view reconstruction task.** We visualize the novel view results generated from the GSO dataset (Downs et al., 2022) (*row 1-3*), or a generated image (*row 4*). Note that for the generated image, we use Zero123++ (Shi et al., 2023) to generate 6 views, which are used as input for InstantMesh (Xu et al., 2024a) and GTR (Ours).

## Abstract

We propose a novel approach for 3D mesh reconstruction from multi-view images. We improve upon the large reconstruction model LRM (Hong et al., 2024) that use a transformer-based triplane generator and a Neural Radiance Field (NeRF) model trained on multi-view images. We introduce three key components to significantly enhance the 3D reconstruction quality. First of all, we examine the original LRM architecture and find several shortcomings. Subsequently, we introduce respective modifications to the LRM architecture, which lead to improved multi-view image representation and more computationally efficient training. Second, in order to improve geometry reconstruction and enable supervision at full image resolution, we extract meshes from the NeRF in a differentiable manner and fine-tune the NeRF model through mesh rendering. These modifications allow us to achieve state-of-the-art performance on both 2D and 3D evaluation metrics on Google Scanned Objects (GSO) dataset (Downs et al., 2022) and OmniObject3D dataset (Wu et al., 2023). Finally, we introduce a lightweight per-instance texture refinement procedure to better reconstruct complex textures, such as text and portraits on

assets. To address this, we introduce a lightweight per-instance texture refinement procedure. This procedure fine-tunes the triplane representation and the NeRF's color estimation model on the mesh surface using the input multi-view images in just 4 seconds. This refinement achieves faithful reconstruction of complex textures. Additionally, our approach enables various downstream applications, including text/image-to-3D generation.

# 1 INTRODUCTION

The task of generating 3D assets from text or images has wide applications in digital content generation and virtual reality (VR) (Lin et al., 2023a;b). Because of the scarcity of 3D data, efforts have been made to leverage pre-trained large-scale text-to-image diffusion models. Some works learn 3D asset generation from the pre-trained image diffusion models via score distillation techniques (Poole et al., 2023; Wang et al., 2023b;a; Chen et al., 2023; Lin et al., 2023a; Zhu & Zhuang, 2023; Qian et al., 2024), which, however, require from several minutes to several hours for a single asset, since the optimization algorithm requires many iterations. Other works propose to fine-tune image diffusion models into a multi-view image diffusion model (Liu et al., 2023; Shi et al., 2024; Liu et al., 2024; Long et al., 2023) using 3D asset datasets (Deitke et al., 2023). These multi-view images serve as an intermediate 3D representation, which is taken as input by large 3D reconstruction models (LRMs) (Hong et al., 2024; Li et al., 2024; Xu et al., 2024c; Wang et al., 2024; Tang et al., 2024) for asset generation. However, these previous approaches struggle to reconstruct faithful textures and high-quality geometry when using the Marching Cube (MC) algorithm (Lorensen & Cline, 1998). Moreover, after extracting meshes using MC, the texture quality degrades even further.

In contrast, we start developing the feed-forward mesh generation model by carefully examining the standard LRM architecture. First, we observed that DINO features tend to discard high-frequency image details, which are important for the precise reconstruction, from the input images. Thus we replace the pre-trained DINO transformer (Caron et al., 2021) used in previous LRMs (Hong et al., 2024; Li et al., 2024; Xu et al., 2024c; Wang et al., 2024; Tang et al., 2024) with a convolutional encoder for the multi-view images. Moreover, because of the high computation requirements of the transformers, previous methods usually run a transformer at $32^2$ triplane resolution and use a deconvolution to upsample this triplane. However, we noticed that reconstruction from standard LRM often exhibits regular grid artifacts (see Fig. 1). We speculate that the nature of these artifacts is similar to grid-shape artifacts observed in 2D deconvolutional generators (Odena et al., 2016). To address this, we replace all deconvolution layers with Pixelshuffle layers (Shi et al., 2016). Finally, we employ two shallow Multi-layer Perceptrons (MLPs) to separately predict density and colors, which is beneficial for our following fine-tuning stages, which we will explain shortly.

Learning meshes is significantly harder than learning a NeRF. Thus, we begin by training this modified architecture using NeRF volume rendering. With the trained NeRF model in place, we proceed to fine-tune the pipeline using mesh rendering (also known as rasterization). To achieve this, we employ Differentiable Marching Cubes (DiffMC) (Wei et al., 2023) to extract meshes from the NeRF density fields by transferring the densities to a signed distance function (SDF) representation. This enables us to render full-resolution images for supervision. Additionally, we employ a depth loss to guide the geometry extraction. Our feed-forward mesh generation pipeline significantly boosts the quality of the reconstructions compared to the results extracted from NeRFs using MC.

While the feed-forward mesh generation pipeline achieves advanced 3D reconstruction quality, it still faces challenges in accurately reconstructing intricate textures, such as text and complex patterns. To address this limitation and further enhance texture quality, we introduce a straightforward yet highly effective texture refinement procedure. Specifically, we fine-tune both the triplane feature and the color estimation model for each instance using sparse multi-view input images. As mentioned before, the shallow and separate density and color estimation models enable efficient updating of colors for surface points on the extracted meshes. In other words, the heavy triplane generator remains fixed in this texture refinement stage. This enables us to achieve rapid optimization, reaching 5 iterations per second on an A100 GPU. Remarkably, our method achieves faithful texture reconstruction with just 20 steps of fine-tuning on 4-view images, requiring a mere 4 seconds on an A100 GPU.

Our proposed approach enables faithful 3D reconstruction from the multi-view input images, as shown in Fig. 1. We conduct a comprehensive comparison of our method with multiple concurrent

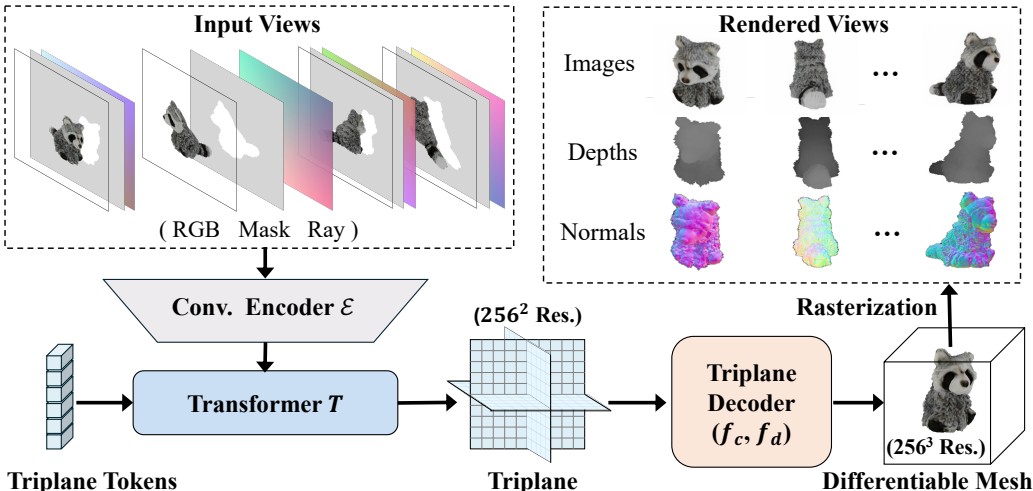

Figure 2: **Overview of our proposed approach for 3D reconstruction.** Our pipeline consists of a convolutional decoder $\mathcal{E}$, a transformer-based triplane generator, $T$, and a NeRF-based triplane decoder that contains two MLPs, $f_c$ and $f_d$, for color and density prediction, respectively. In practice, the triplane resolution is set to 256, and the mesh representation has a grid size resolution of 256.

works (Hong et al., 2024; Xu et al., 2024a; Tang et al., 2024) using the Google Scanned Object (GSO) (Downs et al., 2022) and the OmniObject3D (Wu et al., 2023) datasets, employing various evaluation metrics. For instance, in the 4-view reconstruction task using GSO dataset (Downs et al., 2022), our approach achieves a Peak Signal-to-Noise Ratio (PSNR) of 29.79, a Structural Similarity Index (SSIM) of 0.94 and a Learned Perceptual Image Patch Similarity (LPIPS) score of 0.059.

Extensive experiments show that our feed-forward model achieves superior results compared to the baseline approaches, while our per-instance refinement approach enables further texture improvement on text and complex patterns. See Fig. 1 for visual examples.

We summarize our technical contributions to two key components of multiview-to-3D reconstruction: (1) mesh generation and (2) accurate texture reconstruction. These contributions are outlined as follows:

- We introduce a holistic design for multi-view image to 3D reconstruction to enhance the quality of the generated meshes. This includes modifications to the existing LRM model and fine-tuning the NeRF model with a differentiable mesh representation.

- We present an efficient per-instance texture refinement process, leveraging input images to enhance texture details.

Furthermore, our model can be adapted to various downstream applications, such as text/image-to-3D generation tasks.

## 2 RELATED WORK

**Optimization-based 3D generation** aims to use pre-trained large-scale text-to-image diffusion models Rombach et al. (2022); Saharia et al. (2022) for 3D generation, given the insufficient scale and diversity of existing 3D datasets. To distill 3D knowledge from the text-to-image models, a Score Distillation Sampling (SDS) approach and its variants Poole et al. (2023); Wang et al. (2023a;b); Chen et al. (2023); Zhu & Zhuang (2023) have been proposed. In these methods, noise is added to an image rendered from 3D models like NeRFs and subsequently denoised by a pre-trained text-to-image generative model Rombach et al. (2022). The SDS approach aims to minimize the Kullback-Leibler (KL) divergence between a prior noise distribution and the estimated noise distribution from the text-to-image model. However, these SDS-based methods are time-consuming, usually taking up to hours to generate a single instance. Alternatively, feed-forward 3D generation models have been proposed to achieve faster generation.

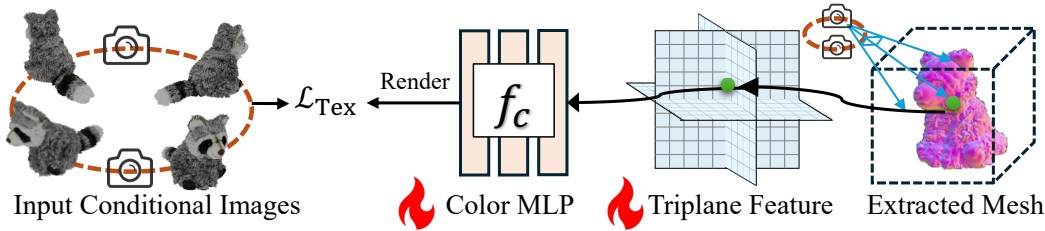

Figure 3: **Texture refinement for extracted meshes.** We refine the texture by fine-tuning the triplane feature of the asset and the color MLP, $f_c$, using the input images. The learnable components are marked as 🔥. We use an L2 loss on the images, defined as $\mathcal{L}_{\text{Tex}}$.

**Feed-forward 3D generation** has gained increasing attention recently due to its speed advantage. Most existing feed-forward 3D generation pipelines consist of two stages: (i) a text prompt (or a single image) to multi-view generation and (ii) multi-view images to 3D shape reconstruction. In the first stage, multi-view images are generated from text or image input using a multi-view generator Liu et al. (2023); Shi et al. (2024); Liu et al. (2024); Long et al. (2023), which are usually fine-tuned from image or video diffusion generation models Rombach et al. (2022); Blattmann et al. (2023). In the second stage, a 3D instance is reconstructed from input multi-view images. In this case, various 3D representations are used such as neural implicit fields Li et al. (2024); Xu et al. (2024c); Hong et al. (2024), Gaussian Splatting Xu et al. (2024b); Tang et al. (2024), or meshes Wang et al. (2024); Xu et al. (2024a). However, existing methods encounter challenges to either faithfully reconstruct textures when using implicit field representations Li et al. (2024); Xu et al. (2024c); Hong et al. (2024), or difficult to extract explicit geometries when using Gaussian Spatting Xu et al. (2024b); Tang et al. (2024). In this work, we focus on the second stage of the pipeline, i.e., reconstructing 3D shapes from multi-view images. To address those challenges, we propose a novel 3D reconstruction approach that improves the 3D quality through geometry and texture refinement.

**Differentiable mesh** is a hybrid 3D representation that combines implicit and explicit surface representations, i.e., SDFs and meshes, and is suitable for 3D optimization. Recent popular representations include DMTet, Flexicubes, Differentiable Marching Cubes (DiffMC) Shen et al. (2021; 2023); Wei et al. (2023). In our work, we chose DiffMC Wei et al. (2023) as it doesn't require additional components beyond our existing model pipeline, in contrast to using DMTet and Flexicubes Shen et al. (2021; 2023), where a deformation and a weight prediction net are required.

**MVS-based 3D reconstruction** aims to generate novel views from sparse-view input images using Multi-view Stereo (MVS) techniques. Classic MVS methods leverage cost volumes Kutulakos & Seitz (2000); Seitz & Dyer (1999), point clouds Stereopsis (2010); Lhuillier & Quan (2005), or depth maps Campbell et al. (2008); Gallup et al. (2007) to learn blending weights for input sparse-view pixels for generating novel views. Recently, learning-based methods Gu et al. (2020); Ma et al. (2021); Wang et al. (2021); Wei et al. (2021); Yi et al. (2020) have been proposed that can generalize to novel scenes. However, these methods require input views to have dense local overlap and struggle to generate 360-degree views of 3D assets. It is even more challenging when input multi-view images are generated without precise pixel-level alignment. Alternatively, in our work, we propose a simple yet effective texture refinement procedure that enables high-quality texture reconstruction from sparse-view input and is robust to synthetic images.

## 3 METHOD

We separate the technical details of our method into three parts. In Sec. 3.1, we explain the modifications made to the existing LRM architecture. In Sec. 3.2, we present a training procedure for our feed-forward mesh generation model. The overview of our feed-forward mesh generation pipeline is illustrated in Fig. 2. Finally, in Sec. 3.3, we introduce our per-instance texture refinement procedure. This procedure is highlighted in Fig. 3.

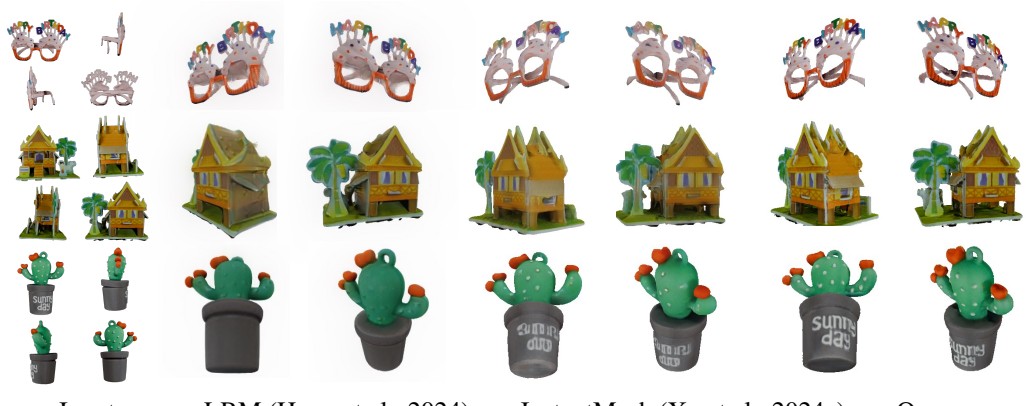

| Input | LRM (Hong et al., 2024) | InstantMesh (Xu et al., 2024a) | Ours |

Figure 4: **Comparison with the baseline methods on OmniObject3D dataset** (Wu et al., 2023). We compare the novel view reconstruction results with the baseline methods. LRM (Hong et al., 2024) takes the front-view image as input while Instantmesh (Xu et al., 2024a) and our method take 4 views.

### 3.1 IMPROVING THE LARGE 3D RECONSTRUCTION ARCHITECTURE

A standard LRM (Hong et al., 2024) architecture consists of an image encoder, a transformer-based triplane generator with a deconvolutional triplane upsampler, and a NeRF-based triplane decoder. In our work, we propose several modifications to the LRM architecture outlined next.

**Convolutional image encoder** $\mathcal{E}$. LRM (Hong et al., 2024; Li et al., 2024; Xu et al., 2024c) models typically utilize pre-trained transformer network DiNO ViT (Caron et al., 2021). However, we observe that since DiNO ViT was designed for semantic understanding, it tends to ignore some local details irrelevant to image semantics but required for accurate reconstruction. Thus we propose to replace the DiNO (Caron et al., 2021) architecture with a convolutional encoder. Since this encoder is trained from scratch along with other components it does not exhibit the bias of pre-trained DiNO. An additional advantage of this encoder is that it does not require any modifications to consume additional inputs useful for 3D reconstruction. To this end, we complement the input images with binary foreground mask and Plücker coordinates for camera rays. We show a comparison of the training process using different encoders in Appendix A.

**Triplane upsampler** $T$. One of the LRM architecture shortcomings is the tendency to generate grid-shaped artifacts. We attribute this issue to the deconvolution operation utilized in a triplane upsampler. Indeed, to reduce the computational requirements, the original LRM proposed to run a transformer-based triplane generator on $32^2$ resolution and later utilize deconvolution operation to upsample the triplane. The deconvolution operation is widely studied in GAN literature, for example, Odena et al. (Odena et al., 2016) show that for 2D generators deconvolutions are the main source of grid-shaped artifacts. To this end, we replace the deconvolution upsampling with a linear layer followed by a pixelshuffle (Shi et al., 2016). This simple modification helps to alleviate grid-shaped texture artifacts.

**NeRF decoders** $f_c, f_d$. Unlike previous LRMs (Hong et al., 2024; Li et al., 2024), we utilize two separate MLPs, defined as $f_c$ and $f_d$, to estimate colors and density, respectively. This modification does not impact performance; however, it serves a more practical purpose. For instance, we can train the color model $f_c$ and freeze $f_d$ when fine-tuning asset texture, or vice versa if fine-tuning asset geometry.

### 3.2 FEED FORWARD MESH GENERATION MODEL

The optimization through mesh representation may pose a significant challenge. Indeed, the gradients for backpropagation through mesh exist only in a small local neighborhood and, thus, convergence heavily depends on accurate initialization. To tackle this, we develop a two-stage training procedure,

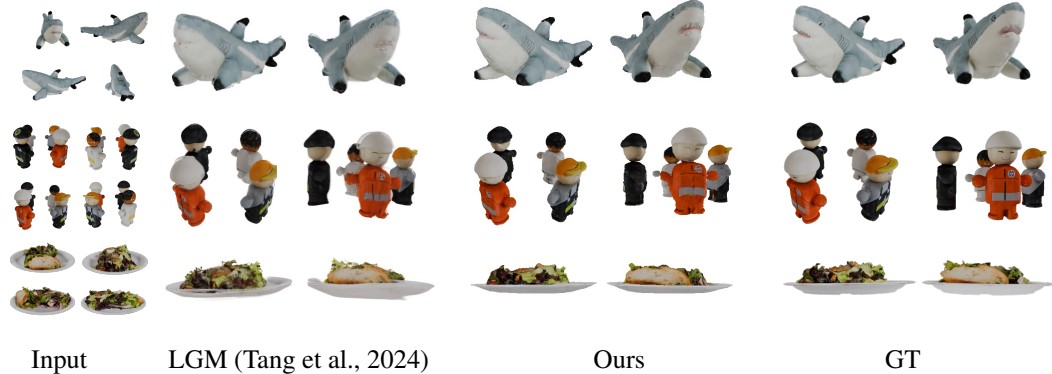

Input         LGM (Tang et al., 2024)         Ours         GT

Figure 5: **Comparison with LGM (Tang et al., 2024) on GSO (Downs et al., 2022) dataset**. Both LGM (Tang et al., 2024) and our method take 4-view images as input and reconstruct novel views.

where in the first stage we utilize the volumetric rendering and optimize NeRF, and, in the second stage, we perform geometry refinement by optimizing through a mesh representation.

**NeRF training stage**. In this stage, we simply optimize our modified architecture (see Sec. 3.1) using a mean squared error (MSE) loss and an LPIPS loss on images, defined as

$$\mathcal{L}_{\text{NeRF}} = \mathcal{L}_{\text{rgb}} + \lambda_{\text{p}} \mathcal{L}_{\text{LPIPS}}, \tag{1}$$

where $\lambda_{\text{p}}$ denotes the loss weight.

**Geometry refinement with NeRF initialization**. In the geometry refinement stage, we fine-tune the entire pipeline using mesh rendering. Specifically, we transfer the density field to an SDF field:

$$\text{sdf} = -(d - s), \tag{2}$$

where $d$ is the estimated density and $s$ is a pre-defined level set for DiffMC, $s = 10$, in practice.

Next, we render images, depths, and masks for training via mesh rendering (a.k.a rasterization). We fine-tune the pipeline using an MSE loss and an LPIPS loss on images, MSE losses on depths, masks, and normal maps. Formally, we define the loss as:

$$\mathcal{L}_{\text{Mesh}} = \mathcal{L}_{\text{rgb}} + \lambda_p \mathcal{L}_{\text{LPIPS}} + \lambda_d \mathcal{L}_{\text{depth}} + \lambda_m \mathcal{L}_{\text{mask}} + + \lambda_n \mathcal{L}_{\text{normal}} \tag{3}$$

where $\lambda_{\text{p}}$, $\lambda_{\text{d}}$, $\lambda_{\text{m}}$ and $\lambda_n$ are the loss weights.

### 3.3 TEXTURE REFINEMENT FOR MESH REPRESENTATIONS

Inspired by previous works that utilize the Gaussian Splatting (Tang et al., 2024; Xu et al., 2024b) representation, where colors of the input images can be easily retained in Gaussian features, we notice a disparity in our pipeline, which lacks this color memorization scheme. Thus, we refine the triplane feature of an asset and the color MLP, $f_c$, using the input multi-view images, $\boldsymbol{I}_{\text{cond}}$, for surface points on the extracted mesh. We illustrate the refinement procedure in Fig. 3. The separated density and color MLPs benefit the texture refinement procedure, as we only fine-tune a single MLP. We use an MSE loss on input images for texture refinement:

$$\mathcal{L}_{\text{Tex}} = \mathcal{L}_2(\boldsymbol{I}_{\text{cond}}, \hat{\boldsymbol{I}}_{\text{cond}}), \tag{4}$$

where $\boldsymbol{I}_{\text{cond}}$ and $\hat{\boldsymbol{I}}_{\text{cond}}$ denote the ground-truth and predicted input images, respectively. Note that at this stage, the image encoder $\mathcal{E}$ and the triplane generator $T$ are fixed and used to generate the initial triplane features of assets. The density estimation MLP, $f_d$, is also fixed and used to extract meshes.

Table 1: **Quantitative comparison on GSO dataset** (Downs et al., 2022).

| Method | PSNR (↑) | SSIM (↑) | LPIPS (↓) | CD (↓) | IoU (↑) |
|---|---|---|---|---|---|
| LRM (Hong et al., 2024) | 20.446 | 0.904 | 0.126 | 6.383 | 0.352 |
| SV3D (Voleti et al., 2024) | 22.098 | 0.898 | 0.119 | 1.770 | 0.682 |
| CRM (Wang et al., 2024) | 22.195 | 0.891 | 0.150 | 1.252 | 0.617 |
| LGM (Tang et al., 2024) | 25.227 | 0.925 | 0.066 | 1.373 | 0.601 |
| InstantMesh (NeRF) (Xu et al., 2024a) | 24.740 | 0.923 | 0.080 | 1.101 | 0.635 |
| InstantMesh (Mesh) (Xu et al., 2024a) | 24.444 | 0.920 | 0.08 | 1.115 | 0.645 |
| Ours (Feed-forward) | 28.673 | 0.946 | 0.055 | **0.740** | **0.708** |
| Ours (Tex. refine) | **29.788** | **0.960** | **0.047** | | |

## 4 EXPERIMENTS

In Sec. 4.1, we explain training datasets, implementation details, and evaluation with the baseline methods. We present both quantitative and qualitative results in Sec. 4.2 and the ablation study in Sec. 4.3. Additional implementation details and comparisons are presented in Appendix.

### 4.1 EXPERIMENT SETTINGS

**Dataset**: Our model is trained on a 140k asset dataset, which merges the filtered Objaverse dataset (Deitke et al., 2023) with an internal 3D asset dataset. We filter the high-quality Objaverse dataset to retain 26k superior assets of high quality. We render 32 random views for each training asset. In Appendix A, we also provide additional ablation studies with our model trained solely on the Objaverse dataset containing 100k assets.

**Implementation Details**. In practice, the loss weights are set to $\lambda_p = 0.5, \lambda_d = 0.5, \lambda_m = 1$ and $\lambda_n = 1$. In practice, we found that adding the normal loss can lead to unstable training. Therefore, during the geometry refinement stage, we add the normal loss once the model has stabilized, and we freeze the generator when incorporating the normal loss. Input multi-view images are of 512 resolution. The triplane transformer contains 24 attention blocks with a hidden dimension of 1024. Each attention layer has 16 attention heads and each head has a dimension of 64. During the NeRF training stage, images are rendered at 512 resolution, and the NeRF model is trained using a patch size of $128^2$. We uniformly sample 256 points along each camera ray. The density and color MLPs consist of 3 and 4 layers, respectively, with a hidden size of 512.

In the NeRF training stage, we use an AdamW optimizer with a learning rate $1e - 4$ and a weight decay of 0.05. Cosine scheduling is employed to gradually reduce the learning rate to 0 after 150k training iterations. We use a batch size of 512 on 32 A100 GPUs. For each asset, we randomly choose 4 views as input and another 4 views for supervision. In the geometry refinement stage, we choose a grid size of 256 during mesh extraction using DiffMC. We use another AdamW optimizer with a learning rate $5e - 5$. The batch size is 192 on 32 A100 GPUs. For each asset, we randomly choose 4 views as input and another 8 views for supervision. In the per-instance texture refinement stage, the learning rates for the triplane feature and the color MLP, $f_c$, are 0.15 and $1e - 4$, respectively.

**Evaluation**. We evaluate our method alongside baseline methods, including LRM (Hong et al., 2024), SVD (Voleti et al., 2024), CRM (Wang et al., 2024), InstantMesh (Xu et al., 2024a), and LGM (Tang et al., 2024) using the Google Scanned Objects (GSO) (Downs et al., 2022) and OmniObject3D (Wu et al., 2023) dataset. We use the identical data lists of the GSO and OmniObject3D dataset and render camera orbits as outlined in Instantmesh (Xu et al., 2024a). Specifically, 300 GSO assets and 130 OmniObject3D assets (from 30 classes) are used for evaluation. We render 20 images for each asset in a trigonometric orbiting trajectory, i.e., maintaining uniform azimuths and elevations in $\{-30°, 0°, 30°\}$. We use PSNR, SSIM, and LPIPS as image evaluation metrics, while Chamfer Distance (CD) and mIoU are utilized for 3D geometry evaluation. To evaluate 3D geometry, we follow the mesh processing steps in InstantMesh (Xu et al., 2024a). Specifically, we reposition the generated meshes to the origin and align the coordinate system with the ground-truth meshes. We then rescale all meshes into a $[-1, 1]^3$ cube. We also use Iterative Closest Point (ICP) registration to

Table 2: **Quantitative comparison on OmniObject3D dataset** (Wu et al., 2023).

| Method | PSNR (↑) | SSIM (↑) | LPIPS (↓) | CD (↓) | IoU (↑) |
|---|---|---|---|---|---|
| LRM (Hong et al., 2024) | 18.082 | 0.888 | 0.125 | 4.347 | 0.448 |
| LGM (GS) (Tang et al., 2024) | 22.826 | 0.913 | 0.068 | 0.893 | 0.626 |
| InstantMesh (NeRF) (Xu et al., 2024a) | 22.609 | 0.914 | 0.076 | 0.660 | 0.671 |
| InstantMesh (Mesh) (Xu et al., 2024a) | 22.141 | 0.910 | 0.079 | 0.603 | 0.675 |
| Ours (Feed-forward) | 25.372 | 0.931 | 0.06 | **0.504** | **0.728** |
| Ours (Tex. refine) | **25.400** | **0.937** | **0.059** | | |

align the generated meshes to the ground truth meshes. We sample 16000 points on the asset surface to compute the CD and IoU scores.

## 4.2 RESULTS

**Qualitative evaluation**. We compare our method with baseline approaches using the GSO (Downs et al., 2022) and OmniObject3D (Wu et al., 2023) datasets in Fig.1, Fig.4-5. We observe that our method achieves more faithful texture reconstruction with finer details and more accurate geometry. For instance, in Fig. 1, our model can generate clear text on the first example and complex texture patterns on the second example. In the third example, our model enables the reconstruction of portraits printed on the asset. In Fig.5, we compare our method with LGM (Tang et al., 2024), which uses Gaussian Splatting as a 3D representation. We observe that while LGM (Tang et al., 2024) can generate high-quality textures, it often tends to generate floating Gaussian points in inaccurate regions, even when the input images are ground-truth multi-view images. In Fig. 11, we present a comparison between our results and those of the concurrent MeshLRM approach(Wei et al., 2024). The visual results indicate that our methods produces outcomes comparable to those achieved by MeshLRM. We present additional visual results in Appendix C.

**Quantitative evaluation**. We present the evaluation scores for the GSO (Downs et al., 2022) and OmniObject3D (Wu et al., 2023) dataset in Tab. 1 and Tab. 2, respectively. The CD scores are presented by multiplying a rescale factor of 100. Note that both LGM (Tang et al., 2024) and InstantMesh (Xu et al., 2024a) baselines are concurrent works. For InstantMesh (Xu et al., 2024a), we compare both their results using either neural rendering or mesh rendering. Since LRM (Hong et al., 2024) takes a single image as input, we provide a front-view image to it. Our full approach achieves the best evaluation results in 2D and 3D evaluations. We also present the evaluation scores using our feed-forward model, i.e., without the texture refinement procedure. These results still show significant improvements in texture and geometry quality.

Additionally, the LGM (Tang et al., 2024) method serves as a strong baseline as we directly compare against their generated images rather than rendered images from extracted meshes. We note that the baselines that do not incorporate some explicit geometry representation during training typically yield inferior rendering results when extracted to meshes. However, our method achieves better results than their directly generated images. Furthermore, it takes 1 minute to extract meshes using LGM (Tang et al., 2024) from Gaussian Splatting. In contrast, our method enables to generate meshes within 1 second, plus an additional 4 seconds for texture refinement, which in total is still faster than LGM (Tang et al., 2024) inference.

**Applications**. Our method enables downstream tasks such as text/image-to-3D generation. In this case, we generate multi-view images using pre-trained text-to-image and/or image-to-multiview diffusion models (Rombach et al., 2022; Shi et al., 2023). In practice, we use Zero123++ (Shi et al., 2023) to generate 6-view images as the input for our model. We show the generated results in Fig. 6 and additional results in the Appendix C.

## 4.3 ABLATION STUDY

**Geometry refinement**. In Fig. 7-8, we compare results generated using NeRF+MC and NeRF with geometry refinement. We observe that directly extracting meshes from the NeRF field using MC leads to blurry texture results and a significant drop on 2D evaluation scores. In contrast, after fine-tuning at the geometry refinement stage, the rendered images show improved high-frequency details. The results in Fig. 9 present that the normal loss significantly improves the surface quality.

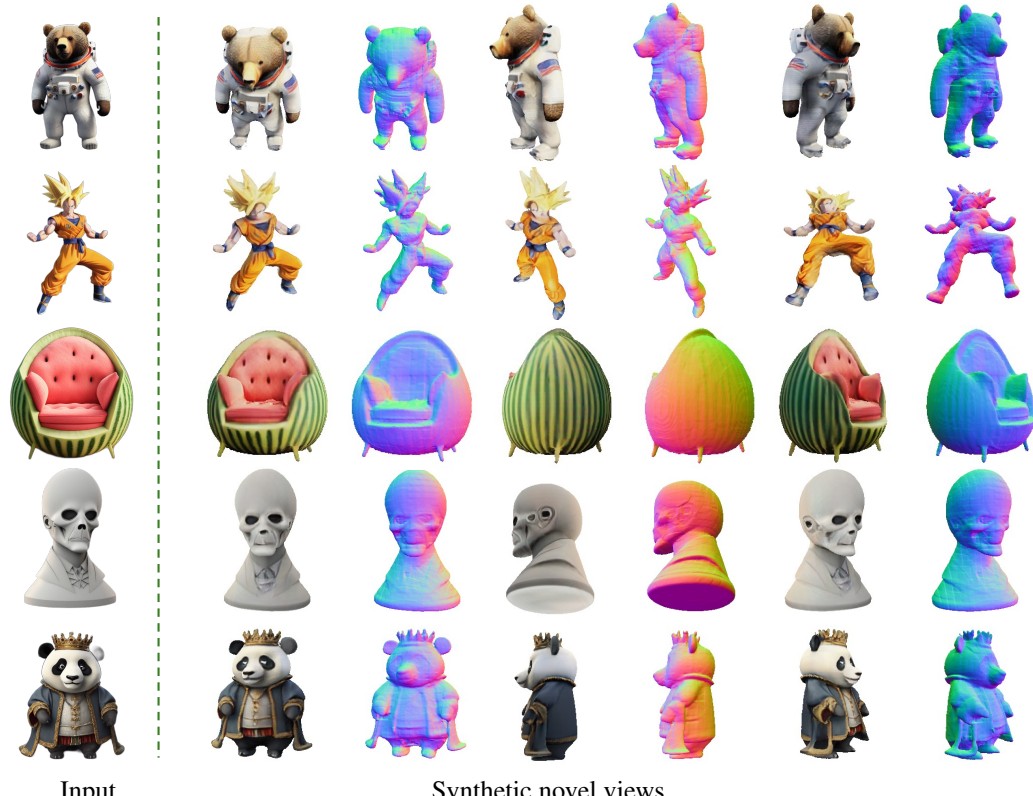

Figure 6: **Image-to-3D generation**. Our method can adapt to the text/image-to-3D generation tasks. We visualize the input image (*column 1*) and generated RGB and normal images from novel views (*column 2-7*).

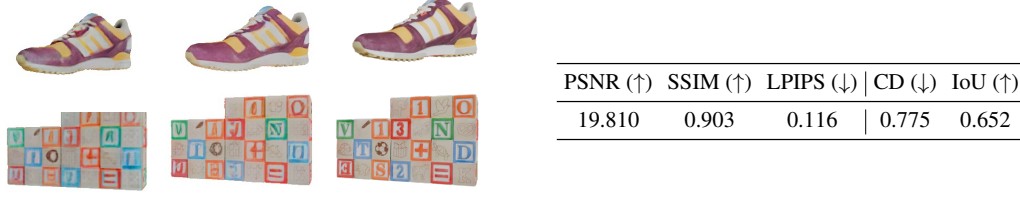

| PSNR (↑) | SSIM (↑) | LPIPS (↓) | CD (↓) | IoU (↑) |
|---|---|---|---|---|
| 19.810 | 0.903 | 0.116 | 0.775 | 0.652 |

NeRF + MC   NeRF+ Geo.   GT

Figure 7: **Ablation study of geometry refinement**. On the left, we visualize the comparison between NeRF+MC, NeRF+Geometry refinement, and the ground-truth images. On the right, we present the evaluation scores using NeRF+MC.

**Texture refinement**. We visualize the novel view mesh-rendering results generated without and with per-instance texture refinement in Fig. 10. The results of the texture refinement appear to have superior detailed textures on mesh surfaces.

## 5 CONCLUSION

In this work, we introduce GTR, a large 3D reconstruction model that takes multi-view images as input. Our approach enables the generation of high-quality meshes with faithful texture reconstruction within seconds. We achieve this through three key contributions: modifications to the current LRM model architecture, the integration of end-to-end geometry refinement with NeRF initialization, and the implementation of a per-instance texture refinement procedure.

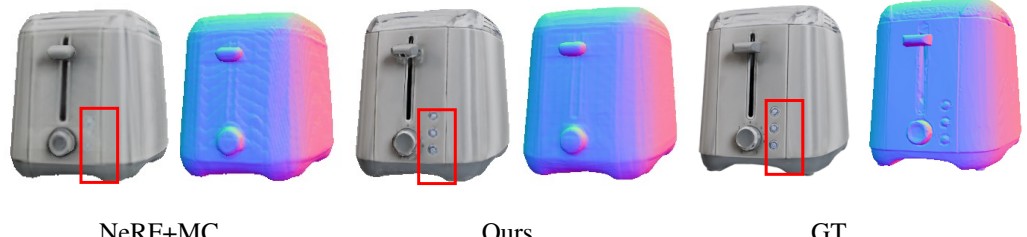

NeRF+MC      Ours      GT

Figure 8: **Additional visual comparison with Marching Cube (MC)**. When the rendering results using MC lack details (marked in red rectangles) and grid artifacts exist on surfaces. In contrast, our geometry refinement achieves better mesh rendering quality and geometry quality than the meshes extracted from NeRFs.

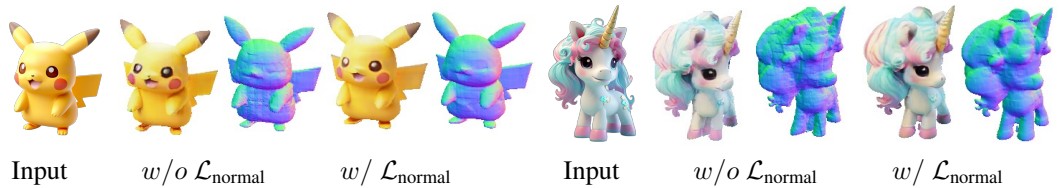

Input    $w/o\ \mathcal{L}_{\text{normal}}$    $w/\ \mathcal{L}_{\text{normal}}$    Input    $w/o\ \mathcal{L}_{\text{normal}}$    $w/\ \mathcal{L}_{\text{normal}}$

Figure 9: **Ablation on the normal loss.** The visual results show that using the normal loss could produce higher-quality surfaces.

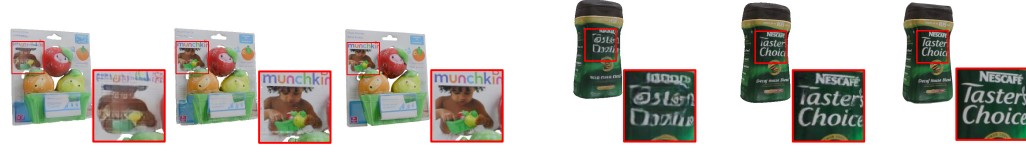

$w/o$ Tex. refine    $w/$ Tex. refine    GT    $w/o$ Tex. refine    $w/$ Tex. refine    GT

Figure 10: **Ablation study of the texture refinement procedure.** We visualize the mesh rendering results without (*column 1, 4*) or with (*column 2, 5*) the texture refinement procedure, and corresponding ground-truth images (*column 3, 6*).

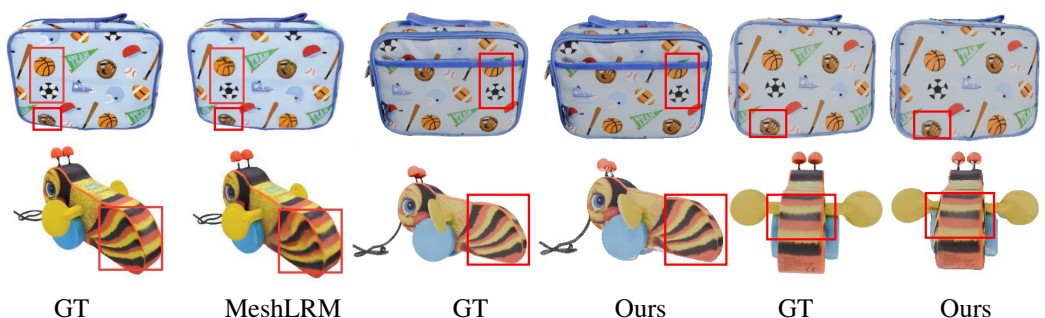

GT     MeshLRM     GT     Ours     GT     Ours

Figure 11: **Visual comparison with MeshLRM** (Wei et al., 2024) on GSO dataset (Downs et al., 2022). Our method generates better (at least comparable) textures with more details than the concurrent work (Wei et al., 2024).

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

We present additional ablation study, limitation, and visual results in the **Appendix manuscript**. Please also refer to our **website demo** in the supplementary material for a comprehensive overview.

## A  ABLATION STUDY

**DiNO encoder**.  We conduct experiments using different encoders.  In Fig. 12 we present the validation PSNR curves during training, when using either our convolutional image encoder or a pre-trained DiNO ViT (Caron et al., 2021). Specifically, our convolutional encoder is a single layer that downsamples input images from 512 to 32. The triplane generator is a self-attention transformer, identical in both settings. We train both models on 8 80G A100 GPUs. We observe that the DiNO experiment did not show improved convergence during the initial iterations. Alternatively, more careful designs could optimize the use of DiNO ViT, which we leave for future study.

**Objaverse training dataset**. In Fig. 12, we also show the training process with a dataset consisting solely of 100k Objaverse (Deitke et al., 2023) images. We did not observe a performance drop in the early stage compared to the other experiments in the figure, which were trained on our mixed dataset.

**Vae encoder**. In Fig.13, we show preliminary results using a pretrained VAE encoder[1] from an SD model (Rombach et al., 2022). To enable the VAE encoder to handle multi-channel input, we separately provide images, masks, and the camera rays to the encoder, then assemble the output features using a convolution layer. Experiments are run on 32 80G A100 GPUs. We observe that using a pretrained VAE encoder leads to better convergence in the early training stage. We attribute this to the good initialization provided by VAEs compared to training the convolutional encoder from scratch.

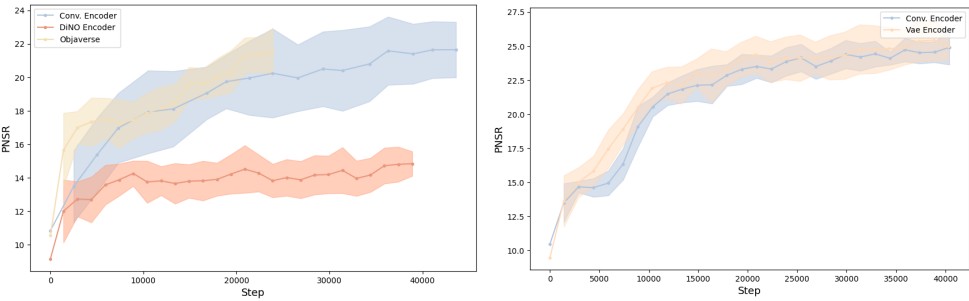

Figure 12: **Ablation study on image encoders** ( Conv. vs DiNO) and dataset.

Figure 13: **Ablation study on image encoders** (Conv. vs VAE).

## B  LIMITATION

Current triplane features at a resolution of 256 are sometimes insufficient to capture fine geometric details.  Meanwhile, alternative representations — explicit (e.g., sparse voxels) or implicit (e.g., vector sets) — offer promising avenues for exploration. Additionally, our approach currently requires camera-conditioned input. It would be more advantageous to develop methods capable of handling unposed input.  Moreover, our work focuses on object-centric assets, leaving the exploration of complex scenes with composite objects as a valuable direction for future research.

## C  ADDITIONAL RESULTS

We show additional results generated by our approach in Fig. 14- 16.

---

[1]In practice, we use the pretrained SD VAE from https://huggingface.co/madebyollin/taesd

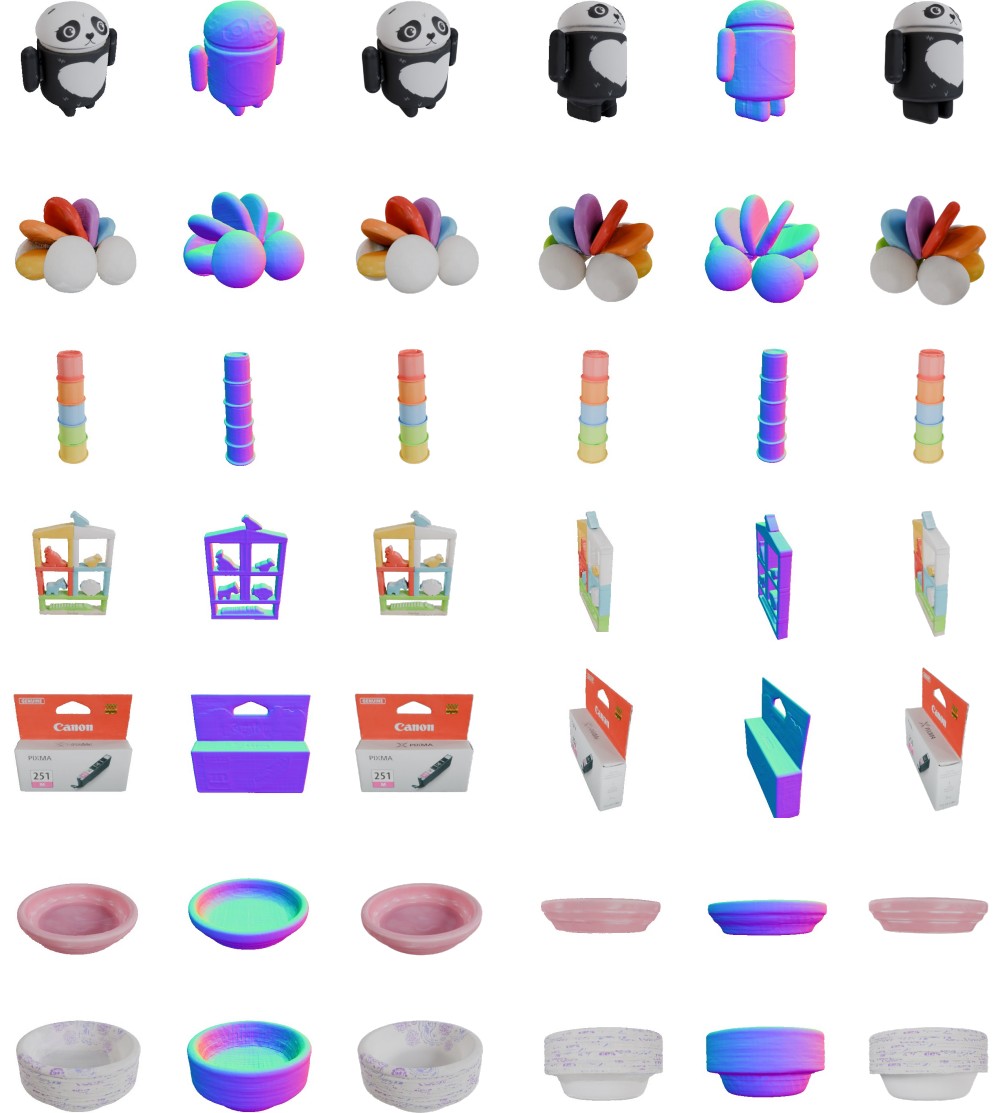

Figure 14: **Additional visual 3D reconstruction results on GSO (Downs et al., 2022) dataset**. The input of our model is 4 orthogonal views. We show the novel view generated RGB images (*column 1, 4*) and normal images (*column 2, 5*), and the ground-truth images (*column 3, 6*).

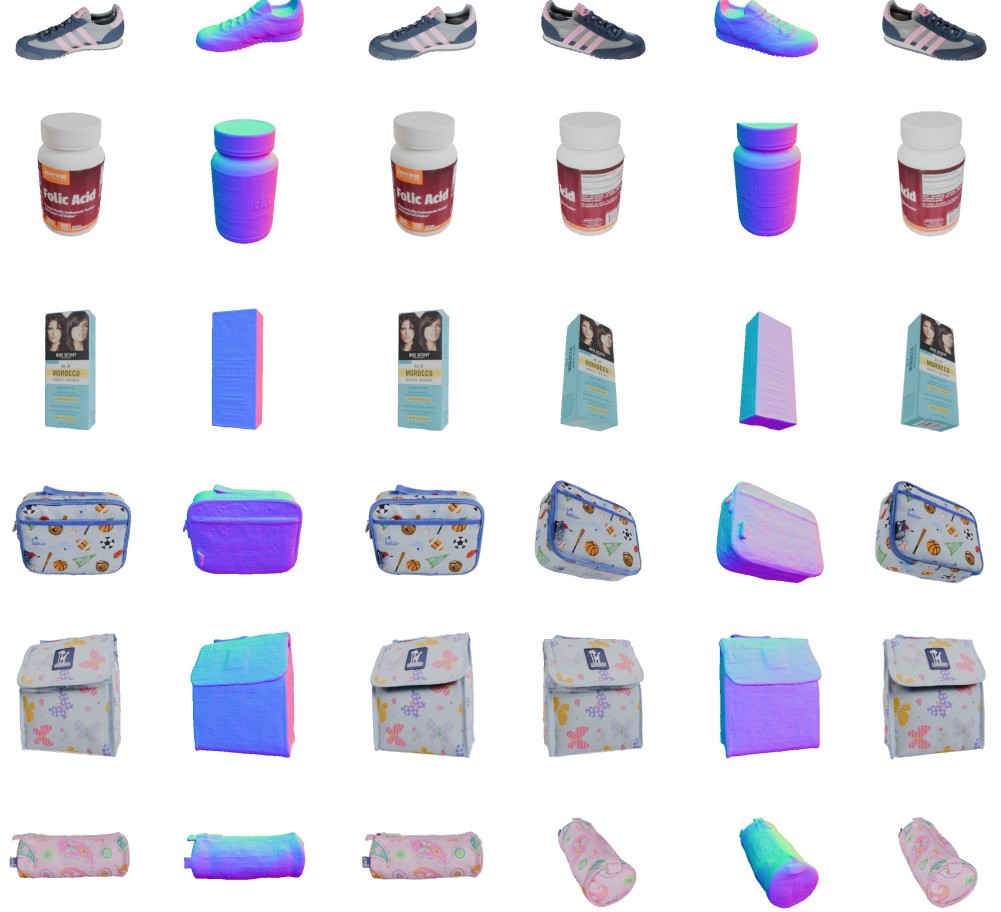

Figure 15: **Additional visual 3D reconstruction results on GSO (Downs et al., 2022) dataset**. The input of our model is 4 orthogonal views. We show the novel view generated RGB images (*column 1, 4*) and normal images (*column 2, 5*), and the ground-truth images (*column 3, 6*).

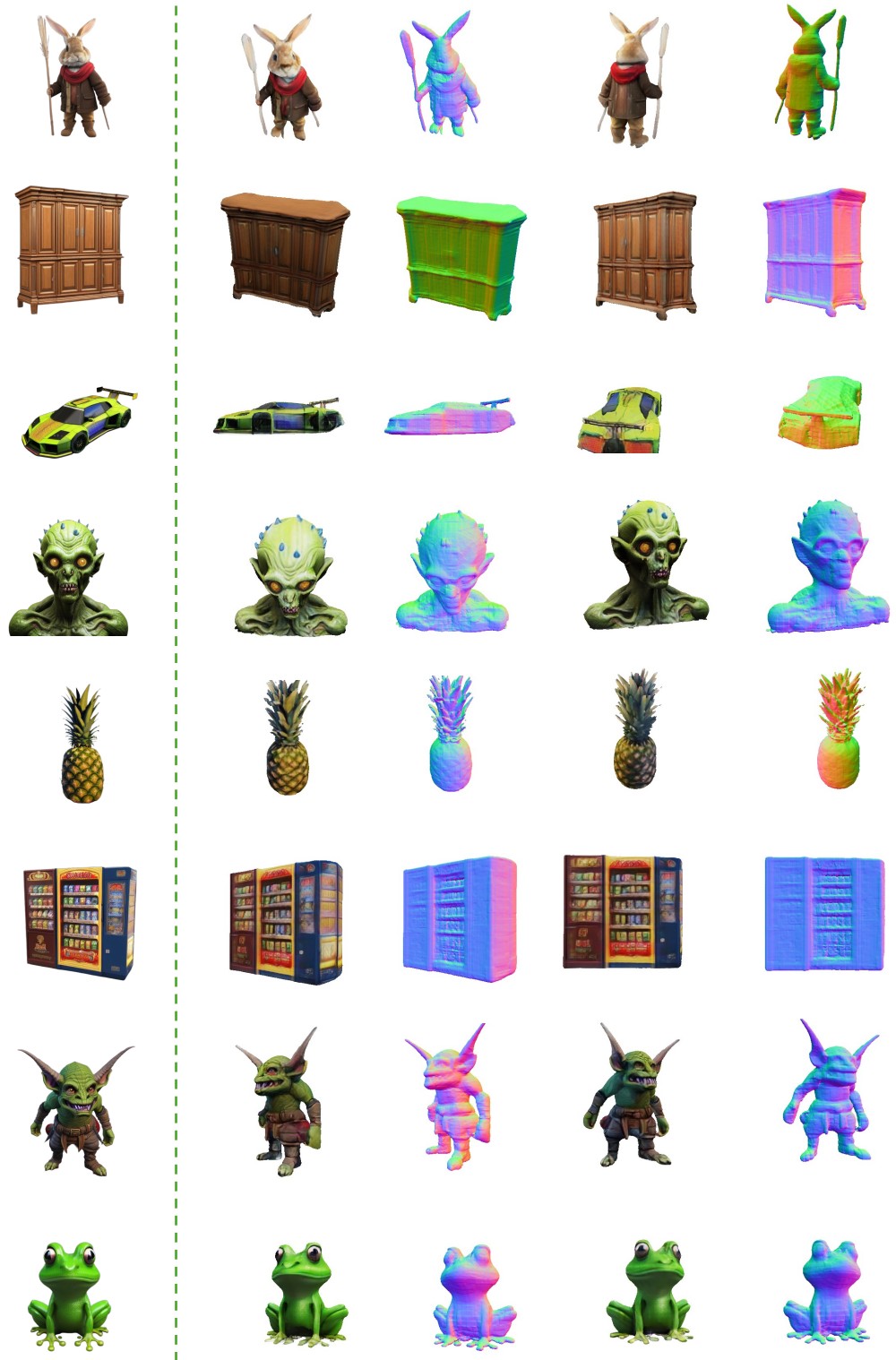

Figure 16: **Additional visual 3D asset generation results**. The input images (*column 1*) are either generated from text using pre-trained text-to-image diffusion models (Rombach et al., 2022) or online generated images.

