# OpenReview forum: "GTR: Improving Large 3D Reconstruction Models through Geometry and Texture Refinement"
_ICLR.cc/2025/Conference — ICLR 2025 Poster_

### Official Review · Reviewer_5Yvj · 2024-11-01

**Soundness:** 3
**Presentation:** 3
**Contribution:** 3
**Rating:** 6
**Confidence:** 4

**Summary:**

This paper builds on the LRM framework with three main improvements: modifications to the LRM model structure, end-to-end geometry refinement with NeRF initialization, and per-instance texture refinement. These enhancements contribute to improved geometry reconstruction and texture quality.

**Strengths:**

1. The proposed methods are well-grounded, and the ablation studies thoroughly support the authors' claims.
2. The visual results appear to achieve superior geometry and texture compared to previous methods.
3. The paper is well-written overall, making it easy to follow.

**Weaknesses:**

1. The approach involves multiple stages, which may require more time compared to the baseline.
2. The paper lacks evaluation metrics for generation quality, such as FID or CLIP scores; including these metrics would strengthen the evaluation.

**Questions:**

Aside from the mentioned weaknesses, I have the following questions:
1. Using DiffMC for geometry could be time-consuming. The paper highlights that the color refinement stage only takes 4 seconds, but it does not mention the time required for the geometry stage. I would like to see a breakdown of the time taken for each stage compared to the baseline.
2. In Section 3.2, where do the depth loss and normal loss supervision come from? If this is a single real-world image, how are depth and normal information obtained?
I would be glad to raise my rating if my concerns are addressed.

---

> ### Author Response · Authors · 2024-11-23
> **Response to Reviewer 5Yvj**
>
> **1. The approach involves multiple stages.**
>
> During inference, the model directly generates a mesh in a feed-forward manner, followed by texture refinement. Although the inference process consists of the two stages, our ablation study in Fig.10 demonstrates the effectiveness of texture refinement. Additionally, we modified the manuscript by adding more comparisons with MeshLRM in Fig.11. It shows how our method facilitates the generation of higher texture details. For future work, we aim to enhance detail generation while simplifying the pipeline and reducing computation time.
>
>
>
> **2. Missing evaluation (FID, CLIP) of generation quality.**
>
> As the model performs a 3D reconstruction task with ground truth novel views of 3D assets available, we use 3D reconstruction metrics to evaluate the quality of the generated outputs. These metrics provide a more accurate reflection of reconstruction accuracy compared to generative metrics.
>
> **3. Time breakdown for each step.**
>
> We provide a detailed breakdown of the time required for generating a single shape on a single 80G A100 GPU. Specifically, generating triplane features from multi-view inputs takes 0.703s, while mesh generation using DiffMC takes 0.126s. At this stage, we obtain a mesh with faces and vertices. We mention that the Marching Cube algorithm is time efficient. Following this, we apply our texture refinement process, which requires an additional 3.518s for 20 iterations of optimization. In total, a shape can be generated within 5s.
>
> | Operation                         | Time     |
> |------------------------------|----------|
> | Triplane Feature Generation  | 0.703s   |
> | Mesh Generation (w/ DiffMC)  | 0.126s   |
> | Texture Refinement           | 3.518s   |
>
>
>
> **4. Source of depth and normal loss supervision.**
>
> We would like to clarify that depth and normal losses are designed for training. The training dataset consists of Obajverse-like 3D assets, where depth and normal information are rendered from the underlying 3D shapes. During inference, the model generates a 3D mesh with the input of multi-view images, where depth and normal information are not required.

---

> > ### Comment · Reviewer_5Yvj · 2024-11-26
> >
> > I think the response adequately addresses my previous questions. I agree with the positive contribution of this paper so I'll maintain my positive ratings. Thanks again for the authors' response.

---

### Official Review · Reviewer_4eVv · 2024-11-02

**Soundness:** 3
**Presentation:** 2
**Contribution:** 2
**Rating:** 6
**Confidence:** 4

**Summary:**

The paper proposed an improved 3D mesh geometry and texture refinement model based on large reconstruction model (LRM) structures.

The authors analyze the limitation of the LRM model and update this with modified LRM architecture, geometry refinement with NeRF initialization, and texture refinement stages.

**Strengths:**

The authors analyze and discuss well about why the previous LRM model has limitations and worse results.

The qualitative results of the paper show improvement over the previous methods, especially for the text.

**Weaknesses:**

There are some typos.

In L. 66, theoptimization → the optimization

In L. 195, Lhuillier & Quan (2005)., → Lhuillier & Quan (2005),

In Fig. 4, LRM takes one front-view image, but GTR (the authors’ method) takes 4 views, which is not a fair comparison. The authors need to use the same front-view image.

The paper lacks discussion and limitations of their method. The authors need to discuss the limitations of their method and the possibilities for further development.

**Questions:**

In L. 91, the authors mention that “this enables us to render full-resolution images for supervision” for the advantage of the DiffMC pipeline. The reviewer wonders why it enables full-resolution images, and the authors need to add the reason; for instance, NeRF MLP needed heavy computation and was hard to use full-resolution, but mesh rasterization had effective computation.

In L. 97, the authors state, “fine-tune both the triplane representation and the color estimation model …”. The reviewer suggests replacing representation with feature, which is consistent with the word used later in the paper and less confusing with the triplane generator.

In L. 243, the reviewer wonders why the authors use Plucker coordinates for their camera ray embedding. Is there any insight or reason?

For the dataset, what is the source of an internal 3D asset dataset, and when retained, 26k superior assets of high quality are used?

In the qualitative results, IoU means 3D mIoU? The authors need to denote 3D if it is 3D metric. In Table 1 and Table 2, the left three columns for 2D metrics and the right two columns for 3D metrics are not specified. Authors need to add these details in the caption.

For the CD metric, the authors need to state their full name (maybe Chamfer Distance) first and use an abbreviation.

---

> ### Author Response · Authors · 2024-11-23
> **Response to Reviewer 4eVv**
>
> **1. Typos and word consistency**
>
> We thank the reviewer for the useful feedback. We fixed typos in the modified version.
>
> **2. Comparison with baselines.**
>
> LRM and SV3D take a single-view image as input, whereas the other baselines that came out more recently, including CRM, LGM, and InstantMesh, use multiple views as input, making them more comparable to our setup.
>
> **3. Illustrate limitations of the work.**
>
> We added limitations to the modified manuscript:
>
> "Current triplane features at a resolution of 256 are sometimes insufficient to capture fine geometric details. Meanwhile, alternative representations— explicit (e.g., sparse voxels) or implicit (e.g., vector sets)—offer promising avenues for exploration. Additionally, our approach currently requires camera-conditioned input. It would be more advantageous to develop methods capable of handling unposed input. Moreover, our work focuses on object-centric assets, leaving the exploration of complex scenes with composite objects as a valuable direction for future research."
>
>
> (Below are responses to the questions)
>
> **4. Full-resolution supervision.**
>
> By employing mesh representation, the surface of an instance can be efficiently detected using the Marching Cubes (MC) algorithm. Subsequently, by leveraging rasterization instead of neural rendering, we eliminate the need for sampling along camera rays.
> Additionally, our SDF and color MLPs are lightweight, consisting of only 3 and 4 layers, respectively. In contrast, MLPs used in NeRF renderings typically have around 8 layers, further highlighting the efficiency of our approach.
>
> **5. Plucker embedding.**
>
> We represent Plücker coordinates as r=(o x d,d), where o and d denote the origin and direction of a camera ray for each pixel, respectively, and x represents the cross-product. In this formulation, both the origin and direction of the camera ray are taken into account, while the embedding remains independent of any specific point along the ray.
>
>
> **6. 3D asset dataset.**
>
> The internal 3D asset dataset was taken outside the company and used during the NeRF training stage, while only the Objaverse dataset was utilized during the geometry refinement stage.
>
> **7. CD and IOU.**
>
> We thank the reviewer for their valuable feedback. In the modified version, we have clarified the text by introducing the full name "Chamfer Distance" (CD) before using its abbreviation. Additionally, we specified that IOU refers to mIOU. Both CD and mIOU are computed in 3D space as mentioned in Sec. 4.1 (Evaluation), where we sample 16k points on the ground truth and generated meshes for evaluation.

---

> > ### Comment · Reviewer_4eVv · 2024-11-26
> >
> > Thank you for the authors' responses. The reviewer decided to raise the score because the authors addressed my concerns well. The reviewer appreciates the detailed explanations of the authors.

---

### Official Review · Reviewer_S5XH · 2024-11-03

**Soundness:** 3
**Presentation:** 3
**Contribution:** 3
**Rating:** 6
**Confidence:** 3

**Summary:**

This work proposes a novel approach for 3D mesh reconstruction from multi-view images. The authors improve upon the large reconstruction model LRM that use a transformer-based triplane generator and a Neural Radiance Field (NeRF) model trained on multi-view images. They introduce three key components to significantly enhance the 3D reconstruction quality. First of all, they examine the original LRM architecture and find several shortcomings. Subsequently, they introduce respective modifications to the LRM architecture, which lead to improved multi-view image representation and more computationally efficient training. Second, in order to improve geometry reconstruction and enable supervision at full image resolution, they extract meshes from the NeRF in a differentiable manner and fine-tune the NeRF model through mesh rendering. The method enables various downstream applications, including text/image-to-3D generation.

**Strengths:**

1. The results are good in terms of both qualitative and quantative result.
2. The experiments are solid. Many baselines are compared.
3. The paper in written very well.

**Weaknesses:**

1. This work has a large improvement in terms of quantative resutls. However, this work differs from previous works only from some incremental improvements in architecture. So what bring this such a large improvement? An ablation study of the proposed tricks will be very appreciated.
2. How long does it take to generate a single shape in total?
3. Will this work be open-sourced?
4. What is the common failure case of this method?
5. In table 1, some baselines has better results than yours (CD, IoU), however, you mark your results in bold font.

**Questions:**

See above

---

> ### Author Response · Authors · 2024-11-23
> **Response to Reviewer S5XH**
>
> **1. Ablation on model architecture.**
>
> We conduct an ablation study to compare upsampling approaches (PixelShuffle vs. Deconv) and different resolutions of the triplanes (32, 44, 128, and 256), as detailed below. All experiments were performed using a single A100 node with 8 GPUs, training the NeRF stage for one day.
>
> | Method                        | Triplane Resolution |  PSNR   |
> |:-----------------------------:|:-------------------:|:-------:|
> | PixelShuffle                 |         32          | 17.373  |
> | PixelShuffle                 |         64          | 18.882  |
> | PixelShuffle                 |        128          | 20.625  |
> | Deconv.                      |        256          | 21.151  |
> | PixelShuffle                 |        256          | 22.281  |
>
>
> Additionally, we conducted ablation of image encoder (DiNO VS Conv.) in the supplementary.
>
> **2. Time cost.**
>
> We provide a detailed breakdown of the time required for generating a single shape on a single 80G A100 GPU. Specifically, generating triplane features from multi-view inputs takes 0.703s, while mesh generation using DiffMC takes 0.126s. At this stage, we obtain a mesh with faces and vertices. Following this, we apply our texture refinement process, which requires an additional 3.518s for 20 iterations of optimization. In total, a shape can be generated within 5s.
>
> | Operation                         | Time     |
> |------------------------------|----------|
> | Triplane Feature Generation  | 0.703s   |
> | Mesh Generation (w/ DiffMC)  | 0.126s   |
> | Texture Refinement           | 3.518s   |
>
>
> **3. Open source.**
>
> To support the academic community, we commit to open-sourcing our work, including the code and pre-trained checkpoints, to facilitate further research and development in 3D generation and downstream applications.
> 4. common failure case of this method.
> We thank the reviewer for the valuable question.We added limitations into the modified manuscript. In our evaluation, when input multi-views are synthetic with pixels that are not perfectly aligned across views, our model has to hallucinate a reasonable shape to handle the inconsistency. As shape reconstruction from synthetic multiviews could be an important application, the generalizability of our shape reconstruction model will be considered for future study.
>
> **5. Table 1.**
>
> We thank the reviewers for the useful feedback. We fixed the typos and updated Table 1.

---

### Official Review · Reviewer_dBU7 · 2024-11-03

**Soundness:** 3
**Presentation:** 3
**Contribution:** 1
**Rating:** 5
**Confidence:** 5

**Summary:**

This paper proposed a feedforward method for extracting meshes from multi-view images
based on large reconstruction models. The paper proposes several improvements in
terms of network architecture, including replacing pretrained DINO encoder with
simple convolution image encoders to capture image details,  and replacing deconvolution
layers with pixelshuffle layers. To extract meshes, the paper proposes to apply
Differentiable Marching Cubes on the density grid. To improve geometry quality, the
paper applies additional losses on depth and normal maps. The paper compares to
previous methods such as LRM and InstantMesh and shows that the proposed method achieves
better geometry reconstructions and renderings.

**Strengths:**

1. The paper proposes several designs that improve the quality of LRM.  The paper performs
ablation studies to validate the effectiveness of such designs.

**Weaknesses:**

1. The overall pipeline of the paper is similar to that of InstantMesh, which also applies
LRM for mesh reconstruction with differentiable iso-surface methods. Both papers apply
losses on depth and normal maps to improve the quality of the geometry.  While the proposed designs
are helpful, they do not provide very significant technical contributions. Recent works
such as MeshLRM, GS-LRM, and LGM have adopted similar strategies to improve the network design
and should be discussed here.

2. The paper does not provide quantitative evaluations on the effectiveness of the network
design choices such as pixelshuffle layers and encoders. Ideally, the numbers should be provided
for the ablation models in Table 1 for better clarity.

3. In the ablation models, is the DiNO encoder frozen or trainable during training? It's not
fully clear why the ViT cannot capture image details considering the existence of residual
connections.

Overall, I am not convinced that the paper presents enough technical novelty to be accepted.
Therefore, I vote for a borderline rejection.

**Questions:**

Please see the weakness.

---

> ### Author Response · Authors · 2024-11-23
> **Response to Reviewer dBu7**
>
> **1. Pipeline design.**
>
> We argue that the "similar overall pipeline" used in previous works did not achieve better results compared to ours, highlighting the effectiveness and contributions of our design improvements. We summarize our contributions in two key aspects:
>
> **Model Improvements:** We identify and implement several effective model modifications based on the "similar overall pipeline." As detailed in Section 3.1 and Section 4.2-4.3, these modifications are both useful and necessary to enhance visual quality. We conducted additional ablation studies for model designs which are shown below.
>
> **Texture Refinement:** We propose a novel texture refinement approach that significantly improves the quality of results, particularly in challenging 3D reconstruction tasks such as reconstructing text and human faces in 3D.
>
> For open-source works, such as LGM, we perform both visual and quantitative comparisons. However, for non-open-source works like MeshLRM, which lack publicly available implementations or official demos, direct comparison is challenging. In such cases, we infer comparisons qualitatively based on reported results.
>
> In Figure 11 of the revised manuscript, we provide additional visual comparisons with MeshLRM. The results on the GSO dataset demonstrate that our proposed texture refinement method achieves detailed and high-quality textures, showcasing its effectiveness.
>
> **2. Ablation on model architecture.**
>
> We conduct an ablation study to compare upsampling approaches (PixelShuffle vs. Deconv) and different resolutions of the triplanes (32, 44, 128, and 256), as detailed below. All experiments were performed using a single A100 node with 8 GPUs, training the NeRF stage for one day.
>
> | Method                        | Triplane Resolution |  PSNR   |
> |:-----------------------------:|:-------------------:|:-------:|
> | PixelShuffle                 |         32          | 17.373  |
> | PixelShuffle                 |         64          | 18.882  |
> | PixelShuffle                 |        128          | 20.625  |
> | Deconv.                      |        256          | 21.151  |
> | PixelShuffle                 |        256          | 22.281  |
>
>
> **3. DiNO encoder vs Conv.**
>
> In our ablation study, the DiNO encoder is trained with the pipeline. We explain this to the inductive bias learned by the DiNO encoder, which makes the model challenging and slow to converge to the new task. While careful design of hyperparameters for DiNO fine-tuning and initialization could improve its performance, in practice, we replace it with convolutional layers, which have fewer parameters and converge more quickly.

---

> > ### Comment · Reviewer_dBU7 · 2024-11-26
> >
> > I thank the authors for the replies. However, I am not fully convinced about the
> > technical contributions:
> >
> > 1. In terms of removing DINO, recent papers such as GS-LRM and MeshLRM have already
> > pointed out this and replaced DINO with just patchify+Linear.
> >
> > 2. Texture refinement is similar in spirit to previous works such as MVSNeRF and
> > IBRNet, which further fine-tunes the feedforward predictions on the input
> > images.

---

> > > ### Author Response · Authors · 2024-11-26
> > > **Response to Reviewer dBU7**
> > >
> > > We acknowledge and appreciate that our work builds upon ideas from the existing literature, as is often the case in iterative scientific progress.
> > >
> > > **1.Image encoder choice**
> > >
> > > We acknowledge that our approach aligns with recent works such as GS-LRM and MeshLRM in replacing DINO with simpler alternatives. However, we note that in these works, the reasoning behind such a replacement is not articulated (GS-LRM), nor do they include ablation studies to evaluate different encoder designs (MeshLRM).
> > >
> > > In contrast, our choice of this design was driven by two critical objectives: **reducing memory costs** and **improving convergence speed**, which are essential for scaling and practical usability. To substantiate this, we presented an ablation study in the appendix that validates the effectiveness of this design choice.
> > >
> > > Moreover, we extended our analysis to explore alternative image encoding approaches to further enhance performance. As shown in Figure 13 of our appendix, we investigated the potential of using the VAE encoder from Stable Diffusion, a well-pretrained and efficient architecture. Preliminary results demonstrate promising potential for this encoder as a superior choice, achieving comparable or better performance while further reducing resource demands. This exploration reflects our effort to go beyond existing work by systematically evaluating and incorporating advanced encoder designs.
> > >
> > > We also wish to emphasize that in the final version of our work, along with releasing the codebase, we plan to train and present a fully developed model leveraging the Stable Diffusion VAE encoder. This will provide the community with a robust, pre-trained encoder alternative that offers practical benefits.
> > >
> > > By focusing on these enhancements, we believe our work not only builds upon prior designs but also explains the rationale behind our model choices, contributing additional value by exploring new possibilities and providing practical alternatives that can inspire further research.
> > >
> > >
> > > **2. Texture Refinement**
> > >
> > > **MVSNeRF**: MVSNeRF requires fine-tuning on dense images captured for a scene, involving 15 minutes of optimization over 10,000 iterations. We quote the MVSNeRF paper: “fine-tune our neural reconstruction for a short period of 15 minutes (10k iterations).”
> > >
> > > **IBRNet**: Similarly, in IBRNet, while the paper does not explicitly mention the time required, optimizing the entire model—including a UNet image encoder, a NeRF MLP, and a ray transformer—is both time- and resource-intensive.
> > >
> > > In contrast, our texture refinement process is designed to be lightweight. Specifically, we fix the mesh geometry and only refine the textures on mesh surfaces using input from four views. In this setup, only a shadow color MLP and a triplane feature are fine-tuned. This requires just 20 iterations and completes within 4 seconds, making our approach significantly more efficient.
> > >
> > > We thank the reviewer for pointing out related works on per-instance optimization. We will include these references in the related work section of our manuscript.

---

> ### Author Response · Authors · 2024-12-02
> **Follow up with Reviewer dBU7**
>
> Dear Reviewer dBU7,
>
> Thanks again for your constructive and insightful feedback to strengthen our work. As the rebuttal period is ending tomorrow, we wonder if our response answers your questions and addresses your concerns. Any feedback is welcome and much appreciated!
>
> Best,
>
> Authors

---

### Official Review · Reviewer_GvuK · 2024-11-03

**Soundness:** 3
**Presentation:** 3
**Contribution:** 2
**Rating:** 5
**Confidence:** 5

**Summary:**

This paper presents GTR, a model for improved 3D mesh reconstruction from multi-view images, which refines both geometry and texture to achieve state-of-the-art quality. It first finds several shortcomings of previous large reconstruction models (LRM) and introduces respective modifications to the LRM architecture, achieving improved quality and more efficient training. The modifications include using a new convolution encoder to replace the DiNO ViT used by previous work, replacing the deconvolution upsampling with a linear layer
followed by a pixelshuffle, etc. Additionally, it presents an efficient per-instance texture refinement process, leveraging input images to
enhance texture details. GTR significantly outperforms baseline models on datasets like GSO and OmniObject3D, supporting downstream tasks such as text/image-to-3D generation.

**Strengths:**

1. Rapid Texture Refinement: The per-object texture refinement is lightweight and achieves faithful texture reconstruction requiring a mere 4 seconds on an A100 GPU.

2. Architecture Modifications: By replacing the DINO ViT transformer with a convolutional encoder and deconvolution layers with a linear layer followed by a pixelshuffle, GTR reduces artifacts of results and enhances high-frequency detail. These changes improve both the visual quality and efficiency of the model training.

3. GTR achieves better results on major 3D reconstruction benchmarks including GSO and OmniObject3D datasets, showing clear quantitative and qualitative improvements over baseline models in 2D and 3D evaluation metrics.

**Weaknesses:**

1. Missing baseline: The paper does not compare with some stronger baselines, such as Mesh-LRM, which has released an online demo from its first author before the ICLR submission deadline.

2. The mesh quality is not satisfactory. In the abstract, the authors said this approach was for "3D mesh reconstruction". However, according to the videos in the supplementary, the video quality is not satisfactory. The surfaces have a lot of bumpy and grid-like artifacts.

3. The overall novelty is limited. The texture refinement part is not very attractive and has been explored in many other related 3D generation works, such as One-2-3-45++, Mesh-LRM, etc. The authors put forward some modifications to the original LRM in this paper.  The original team for the LRM paper has also released several follow-ups for the original LRM, including Mesh-LRM and GS-LRM. Mesh-LRN has shown better results than this work and its modifications seem simpler Therefore, the effectiveness of the proposed modifications should be compared with Mesh LRM's modification.

**Questions:**

1. Please compare with Mesh-LRM

2. Considering Mesh-LRM has shown better results, I am curious about the actual effectiveness of this method's modifications to the original LRM, compared with Mesh-LRM's.
For example, Mesh-LRM replaces DINO ViT with a simple patchify operation; it uses simple "Linear &  Unpatchify" to attach the triplane; it also uses separate MLPs but its MLPs have smaller sizes.

---

> ### Author Response · Authors · 2024-11-23
> **Response to Reviewer GvuK**
>
> **1. Comparison with Mesh-LRM.**
>
> For non-open-source works like MeshLRM, which lack publicly available implementations or official demos, direct comparison is challenging. Inferring implementation differences solely from manuscripts is also challenging. In such cases, we rely on qualitative comparisons based on reported results, as shown in Figure 11 of the modified manuscript.
>
> We argue that the texture refinement is useful and it provides advanced results than the baselines. As shown in Figure 11, while some concurrent works demonstrate improved results in certain aspects, they fall short in achieving challenging 3D reconstruction tasks such as reconstructing detailed textures. In contrast, our proposed texture refinement enables us to achieve superior visual quality. Additional results of our method are shown in Figure 10, where our method can generate good texture details on challenging cases such as human faces and text.
>
> **2. Bumpy geometry.**
>
> We acknowledge that our geometry is not flawless, as bumpy surfaces may occasionally occur. We speculate that this issue could be mitigated with larger-scale training and advancements in 3D representations. Improving the quality of geometry will be a focus of future research.
>
> **3. Limited novelty.**
>
> We understand that this field is rapidly evolving, and newly published arXiv papers may influence the reviewers' decisions. However, we have made every effort to compare our work with concurrent studies included in the manuscript, to the best of our knowledge. We sincerely hope that the reviewers will evaluate our contributions in comparison to works that are accessible.

---

> > ### Comment · Reviewer_GvuK · 2024-11-25
> >
> > I appreciate the authors' replies. Here are my further comments.
> >
> >    1. I have said very clearly in my initial comment, that MeshLRM **" has released an online demo from its first author before the ICLR submission deadline."** The demo link is on MeshLRM's official project page. I am not sure why you say MeshLRM is not publicly available when it is clearly accessible. Both methods show single- image to 3D results and use Zero123++ to get multi-view prediction from the single image input. It should be easy to compare mesh generation quality with MeshLRM under that setting.
> >    I never said texture refinement is not useful. My question is it has been explored in many previous works, which have shown it is useful. This limits the novelty of this work. The papers I mentioned are all for 3d reconstruction tasks.
> >
> >    2. The grid-like artifacts are evident in almost every example in Figure 6. These artifacts do not just "occasionally occur. It would be beneficial if the authors discussed this issue in the limitations section.
> >
> >    3. I was questioning the effectiveness of the proposed modifications, considering MeshLRM has shown better results with simpler modifications. To prove the effectiveness of this work's modifications, the authors could conduct ablation studies comparing it with MeshLRM's modifications. These modifications, such as "Linear & Unpatchify" and the use of smaller, separate MLPs, are relatively straightforward to reimplement for ablation studies, irrespective of whether MeshLRM's code has been open-sourced.

---

> > > ### Author Response · Authors · 2024-11-30
> > > **Response to Reviewer GvuK  about Texture Refinement**
> > >
> > > We added comparisons to clarify how previous approaches (MVSNeRF and IBRNet) relate to our method in the context of texture refinement. Additionally, we are open to incorporating comparisons to other related works if suggested by the reviewer.
> > >
> > > **MVSNeRF**: MVSNeRF requires fine-tuning on dense images captured for a scene, involving 15 minutes of optimization over 10,000 iterations. We quote the MVSNeRF paper: “fine-tune our neural reconstruction for a short period of 15 minutes (10k iterations).”
> > >
> > > **IBRNet**: Similarly, in IBRNet, while the paper does not explicitly mention the time required, optimizing the entire model—including a UNet image encoder, a NeRF MLP, and a ray transformer—is both time- and resource-intensive.
> > >
> > > In contrast, our texture refinement process is designed to be lightweight. Specifically, we fix the mesh geometry and only refine the textures on mesh surfaces using input from four views. In this setup, only a shadow color MLP and a triplane feature are fine-tuned. This requires just 20 iterations and completes within 4 seconds, making our approach significantly more efficient.

---

> ### Author Response · Authors · 2024-11-25
> **Response to Reviewer GvuK**
>
> **“1. MeshLRM has released an online demo from its first author."**
>
> We understand the reviewer is referring to the demo available on the project website. We believe there is some misunderstanding regarding the term "unofficial." As stated on the authors' project page, the demo is described as an unofficial demo reproduced in Hillbot. In this context, "unofficial" implies that aspects such as the training data, model implementation, training resources and training process may not have been faithfully reimplemented as described in the original paper. Consequently, the results produced by the demo may differ from those reported in the original work. Therefore, we chose to base our comparisons on the results presented in the original paper, specifically as shown in Fig. 11 of our paper.
>
>
> **2. The grid-like artifacts in Fig.6.**
>
> We thank the reviewer for the valuable feedback. Upon investigation, originally we found that the grid artifact issue is likely caused by the deconvolution operation on triplanes as used in previous works. To address this, we chose to use pixel shuffle layers to mitigate grid artifacts in textures and geometry. An ablation study was conducted to evaluate this change, as presented in the next. While it improving the quality, further experiments on upsampling techniques can be explored to further eliminate the grid artifacts. We will include this analysis in the limitations section and, as part of future work, conduct more extensive experiments on model design to better address these artifacts.
>
> **3. Ablation of model design.**
>
> We thank the reviewer for the valuable feedback. We conduct an ablation study to compare
>
> (1) upsampling approaches (PixelShuffle vs. Deconv) and different resolutions of the triplanes (32, 44, 128, and 256), as detailed below. All experiments were performed using a single A100 node with 8 GPUs, training the NeRF stage for one day.
>
> | Method                        | Triplane Resolution |  PSNR   |
> |:-----------------------------:|:-------------------:|:-------:|
> | PixelShuffle                 |         32          | 17.373  |
> | PixelShuffle                 |         64          | 18.882  |
> | PixelShuffle                 |        128          | 20.625  |
> | Deconv.                      |        256          | 21.151  |
> | PixelShuffle                 |        256          | 22.281  |
>
>
> (2) DiNO encoder vs Conv.
>
> In the appendix, we present an ablation study on DiNO encoder and Convolution layers. The DiNO encoder is pre-trained, and in our pipeline, it is fined-tuned with our model. In this ablation, we notice that the Convolutional layers shows faster convergence that DiNO. We explain this to the inductive bias learned by the DiNO encoder, which makes the model challenging and slow to converge to the new task. While careful design of hyperparameters for DiNO fine-tuning and initialization could improve its performance, in practice, we replace it with convolutional layers, which have fewer parameters and converge more quickly.

---

> > ### Comment · Reviewer_GvuK · 2024-12-01
> >
> > Thanks for your response. Here are further comments:
> >
> > 1. The fact that MeshLRM uses the word "unofficial" for the demo is most likely just because it was finished by Adobe but the released model was trained using the resources from Hillbot. There may be industry restrictions making them have to use the word "unofficial". But the demo link is placed on the official project page, and it's finished by the first author of MeshLRM. This means the authors of MeshLRM have acknowledged the implementation of the "unofficial" demo is correct. Therefore, it is still reasonable to compare it by using its demo. I appreciate the authors included some comparisons in Fig.11, but I feel it makes more sense to compare with its results in its Fig.6 because in Fig.11 of this paper, it seems that two models also have different image inputs (the color brightness is different, and maybe the camera poses are also different), which makes the comparison potentially unfair for either method.
> >
> > 2. Thanks for your explanation. I agree this explanation should be included in the limitation section
> >
> > 3. The authors have shown simpler Convolution layers are more effective the DINO. I found your convolutional layer is just a single layer, which is almost equivalent to MeshLRM's "Linear & Unpatchify". I am wondering what the difference is here. If they are equivalent, the authors should not claim this point as the contribution of this paper, considering MeshLRM released its paper in April.

---

> ### Author Response · Authors · 2024-12-01
> **Response to Reviewer GvuK**
>
> We appreciate Reviewer GvuK's dedication to the review process and their valuable feedback.
>
> **1. Comparison Using the Public Demo**
>
> We thank Reviewer GvuK for their comment, and we believe we have reached a consensus that the demo is reproduced, which introduces potential ambiguity regarding the data, implementation, training strategy, etc. Consequently, **it is unclear whether the demo outperforms, matches, or underperforms** the model described in the paper.
>
> Following Reviewer GvuK’s suggestion, we evaluated the public MeshLRM demo. We observed that **our results appear qualitatively superior to those provided by the demo.** Specifically, when testing the examples shown in Figure 6 of MeshLRM, we noted several issues with their demo outputs: the fox and tiger examples exhibit blurry colors, while the monkey example displays visibly distorted legs.
>
> Furthermore, we also noticed that **the demo results  seem to differ as well from those presented in their paper, raising questions about the consistency between the demo and the model described in their submission**. Similar concerns have also been noted in the ICLR reviews of MeshLRM, where discrepancies between the demo and paper results were highlighted.
>
> We noticed that the 3rd follow-up comment by the reviewer regarding the demo was not available to us while the manuscript was still editable. However, we would be happy to include visual comparisons to the demo results in the final version to further clarify our evaluation. Nonetheless, we have made every effort to provide fair and thorough comparisons based on the information accessible to us.
>
> **2. Convolution vs Linear & Unpatchify**
>
> - “Linear & Unpatchify”?
>
>  In MeshLRM, “Linear & Unpatchify” is used for upsampling triplane features. As stated in the MeshLRM paper: “Lastly, each triplane token is unprojected with a linear layer and further unpatchified to 8 × 8 triplane features via reshaping. This operator is logically the same as a shared 2D-deconvolution with stride 8 and kernel 8 on each plane.”
>
> In our case, we did not use deconvolution layers for triplane upsampling, as highlighted in the ablation comparisons provided in our last response.
>
>
> - “Patchify & Linear”
>
> We assume the reviewer might be referring to the “Patchify & Linear” operation illustrated in Figure 2 of the MeshLRM paper for extracting image features. However, MeshLRM does not provide detailed implementation specifics for this operation, leaving room for interpretation. It is possible that these two operations could be equivalent theoretically, but without additional details, such equivalence remains speculative.
>
> We fully understand and agree reviewer’s concern and want to clarify that we do not present replacing the encoder as a novel or distinct contribution. Rather, we consider our systematic model modifications as a cohesive contribution, supported by detailed ablation studies that demonstrate the effectiveness of each individual component.

---

> > ### Comment · Reviewer_GvuK · 2024-12-02
> >
> > I appreciate your responses.
> > 1. Thanks for your responses regarding comparing with MeshLRM. I understand the tricky parts when comparing with MeshLRM now. I hope to see the author include visual comparisons to the demo results in the final version to further clarify their evaluation.
> >
> > 2. I apologize for making a TYPO in my previous reply, Yes, I am referring to the “Patchify & Linear”. Based on MeshLRM's paper, it follows Vit and "split the feature map into non-overlapping patches, and linearly transform them to input our transformer." If your single-layer CNN encoder doesn't have a special design, I feel they are very likely to be equivalent. So it's better to mention this in the final revised paper.
> >
> > The authors have solved some of my concerns and I'm a little torn about this work now. If all other reviewers support the acceptance, I will support it, too.

---

### Meta-Review · Area_Chair_Keo8 · 2024-12-14

**Metareview:**

The authors present a large 3D reconstruction model for objects that takes one or multiple images and produces a textured mesh. The model is evaluated against several baseline and convincingly shows superior results.

The paper received mixed borderline reviews of 6/6/6/5/5 after the discussion phase. The reviewers appreciate the high quality results and good evaluations but noticed the limited technical contribution of this work.

After reading the paper as well, I come to similar conclusions. The technical contribution in this work is indeed limited, making a set of smaller adjustments to an existing model. However, the work also addresses a very relevant task that many people are interested in and clear improvements like shown here can have significant impact. The authors also committed to code and model release, which is necessary for this work to have relevant impact.

This paper is truly borderline, due to the above reasons. After some careful consideration I decided to recommend to accept it and would strongly suggest that the authors follow up on their commitment on model and code release.

**Additional Comments On Reviewer Discussion:**

The reviewers are concerned with respect to technical novelty. Pre-rebuttal, they also raised concerns regarding missing comparisons against MeshLRM and missing ablation studies. The authors provided additional comparisons and ablation studies, which addressed many of the given concerns. Specifically, they also provided a comparison against the demo of MeshLRM, as requested by reviewer GvuK, which showcases some examples where GTR provides slightly better results than MeshLRM. I think the comparison is sufficient, given that the authors did not have access to MeshLRMs code.

---

### Decision · Program_Chairs · 2025-01-22

Accept (Poster)